# Balancing Efficiency and Fairness in On-Demand Ridesourcing

**Nixie S. Lesmana**[*]
nixiesap001@e.ntu.edu.sg

**Xuan Zhang**[†]
xuan6@illinois.edu

**Xiaohui Bei**[*]
xhbei@ntu.edu.sg

## Abstract

We investigate the problem of assigning trip requests to available vehicles in on-demand ridesourcing. Much of the literature has focused on maximizing the total value of served requests, achieving efficiency on the passengers' side. However, such solutions may result in some drivers being assigned to insufficient or undesired trips, therefore losing fairness from the drivers' perspective.

In this paper, we focus on both the system efficiency and the fairness among drivers and quantitatively analyze the tradeoffs between these two objectives. In particular, we give an explicit answer to the question of whether there always exists an assignment that achieves any target efficiency and fairness. We also propose a simple reassignment algorithm that can achieve any selected tradeoff. Finally, we demonstrate the effectiveness of the algorithms through extensive experiments on real-world datasets.

## 1 Introduction

Ridesourcing refers to a mode of transportation that connects private car drivers with passengers via mobile devices and applications. Recent advances in technology provide the opportunity for ridesourcing platforms to dynamically match drivers and passengers in real time. This new generation of ridesourcing has the potential to significantly increase the efficiency of urban transportation systems, consequently reducing congestion and pollution [23]. In most on-demand ridesourcing platforms, private-hire car drivers are not allowed to pick up passengers who hail them on the streets, but can only take booking requests assigned by the platform. One key function of these platforms is thus to automatically assign potential passengers to active drivers. The development of an efficient real-time demand assignment algorithm is central to the concept and to the success of a ridesourcing enterprise.

Research into real-time ridesourcing has often focused on developing algorithms for optimal assignment of sets of requests to drivers [1, 33, 24]. In these studies, the common objective is to minimize the total waiting time for passengers and maximize the service rate, achieving efficiency on the passengers' side. Admittedly, customer satisfaction should be the main goal in any service industry. However, in the ridesourcing domain, the role of drivers is as important as that of passengers in terms of sustaining the business. Drivers have preferences that might not align with those of the passengers that are optimized by the algorithm. A centralized algorithm that only focuses on system efficiency will inevitably result in some drivers being assigned to insufficient or undesired trips. Leaving the system as it is would affect the sustainability of the ridesourcing business model in the long run, as unsatisfied drivers will not renew their memberships and new drivers will be deterred from signing up. Therefore, fairness on the drivers' side should be assessed more carefully and should receive more attention.

---
[*]Department of Mathematical Sciences, Nanyang Technological University, Singapore 637371

[†]Department of Industrial and Enterprise Systems Engineering, University of Illinois at Urbana-Champaign, Champaign, IL 61801-3080

In this paper, we study the batch request-vehicle assignment problem with a focus on both efficiency and fairness and address the problem of assigning requests to vehicles that account for the natural tension between these two objectives. In the basic setting, we consider a fleet $\mathcal{V}$ of available vehicles and a set $\mathcal{R}$ of ride requests. Assignment constraints are captured by a bipartite graph $G = (\mathcal{V}, \mathcal{R}, E)$, such that edge $\{v, r\} \in E$ iff vehicle $v$ can be assigned to serve request $r$ under the specified constraints. Our goal is to design an assignment algorithm that matches each vehicle to at most one request, such that both efficiency and fairness are optimized. For efficiency, we adopt the utilitarian criterion, defined as the sum of values of all served requests. This is one of the most axiomatically justified measures of efficiency; it has been studied extensively in the literature and employed as a natural metric for practical applications. For fairness, we adopt the max-min fairness criterion that emphasizes the maximization of the least value that a vehicle obtains. This criterion is built on the Rawlsian egalitarian justice [29] and is well-recognized in various application domains (see more discussions in [34]).

Efficiency and fairness are often competing objectives such that in most cases, the optimum of both cannot be achieved simultaneously. Thus, we are naturally led to the question of how to reconcile system efficiency and drivers' fairness in a ridesourcing assignment. For instance, a central decision maker may choose a mild strategy that balances the tension between efficiency and fairness in normal traffic conditions. However, as demand increases and traffic conditions worsen, it may be desirable to move to a strategy that puts higher attention to efficiency to quickly serve the waiting passengers. Note that such changes are two-way (i.e., efficiency or fairness-oriented), gradual, and dependent on dynamically changing demand conditions. Therefore, to provide better managerial flexibilities to decision makers, we need to provide a full set of candidate allocation solutions and characterize the trade-offs inherent in applying these concepts.

In general, through this paper, we aim to address the following question:

*Given any problem instance and any required fairness threshold, how do we find a request-vehicle assignment that meets the fairness threshold while also has sufficiently good system efficiency?*

## 1.1 Our Contributions

Our contributions can be summarized as follows.

- We answer the above generic question with an efficient algorithm REASSIGN. Our algorithm takes any desired fairness threshold as a parameter, and through a surprisingly simple procedure, computes an assignment that satisfies the fairness threshold and has provably good efficiency.

  Our constructive answer to the question above provides extra managerial flexibilities to decision makers with various needs. For example, in the aforementioned traffic scenario, the requirements for efficiency and fairness vary at different stages. By applying our algorithm, one can simply set a specific fairness requirement as the input of the algorithm to generate an output assignment with the required efficiency and fairness.

- We further show that the efficiency-fairness tradeoff guaranteed by our algorithm is provably optimal. That is, we prove that for any target efficiency and fairness that go beyond our guarantee, there exist a problem instance in which no assignment can achieve both targets simultaneously.

- Finally, we demonstrate the performance of our algorithm in a case study that considers taxi assignment with real taxi data from New York City. Our experiment results show that in practical scenarios, algorithm REASSIGN is able to significantly improve the fairness of the assignment with almost no loss on system efficiency.

## 1.2 Related Works

The problem of vehicle-request assignment in ridesourcing has been studied extensively. Several works have focused on real-time assignment using different approaches, such as greedy match [22], collaborative dispatch [32, 35, 25], planning and learning framework [33], and receding horizon control approach [27].

When requests do not arrive in real-time but are given beforehand, the problem is known as the *Dial-a-Ride Problem (DARP)* [12, 28]. Many variants of the dial-a-ride problem were proposed depending on the specific applications [11, 20, 31, 15, 13, 3, 8].

The idea that a fairness criterion may affect efficiency in resource allocation problems has been explored in numerous contexts and in a variety of models, such as multiportfolio optimization [18], cake cutting [10, 4, 7], load balancing in job scheduling [21, 2, 16], and bandwidth allocation [26, 17, 9]. Bertsimas et al. [5, 6] has specifically studied fairness and the corresponding efficiency loss in a general divisible resource allocation framework and applied their results to a case study in the context of air traffic management. However, none of these works can easily incorporate additional waiting time and pick-up distance constraints because these are unique to the ridesourcing problem. To the best of our knowledge, the efficiency-fairness trade-off in the domain of ridesourcing has not been addressed in the literature.

## 2 Preliminaries

We consider the following bipartite matching problem that models the batch assignment of a set of requests to a set of available vehicles in on-demand ridesourcing. Let us define a bipartite graph $G = (\mathcal{V}, \mathcal{R}, E)$, with $\mathcal{V} = \{v_1, v_2, \ldots, v_n\}$ the set of $n$ available vehicles, $\mathcal{R} = \{r_1, r_2, \ldots, r_m\}$ the set of $m$ requests, and $E \subseteq \{\{v, r\} : v \in \mathcal{V}, r \in \mathcal{R}\}$ the set of weighted edges such that $\{v, r\} \in E$ iff the request $r$ can be served by the vehicle $v$ under some specified assignment constraints[3]. A *utility* $u_{vr}$ is the weight associated to each edge $\{v, r\}$ and defined as the sum of *trip utility* $w_{vr}$ (i.e. the profit vehicle $v$ could obtain by serving request $r$) and *historical utility* $h_v$ (i.e. the total utility $v$ has obtained in preceding assignment periods).

We define *trip value* $\tau_r$ as the length of the trip or the shortest time needed to travel from $r$'s pickup to its dropoff location and *trip cost* $\iota_{vr}$ as the cruising time of $v$ induced by serving $r$. Then, we set our *trip utility* function $w_{vr} = c\tau_r - \iota_{vr}$ with $c$ being a constant to balance the value-cost effect.

Next, we will introduce $\Delta$, a parameter that proves to be critical for our tradeoff analysis in Section 3. Formally,

$$\Delta := \max_{r \in \mathcal{R}} \max_{\{v,r\},\{v',r\} \in E} |w_{vr} - w_{v'r}| \tag{1}$$

or the maximum *trip utility* difference across all pairs of edges corresponding to the same request, across all requests.

*Remarks.* It is important to note that by our definition of *trip utility* $w_{vr}$, the same request $r$ contributes the same value $\tau_r$ to this trip utility when matched with any vehicle. Thus, $\Delta$ directly translates to the maximum difference in vehicle cruising time; it is easy to check that this is bounded above by some assignment constraints that we set, e.g. request waiting time constraint.

We now refer to a setting with graph $G = (\mathcal{V}, \mathcal{R}, E)$, a set of *trip utilities* $\{w_{vr}\}_{\{v,r\} \in E}$ and *historical utilities* $\{h_v\}_{v \in \mathcal{V}}$ as an *instance* $\mathcal{I}$.

Given an instance $\mathcal{I}$, our goal is to find an *assignment* $M$ that assigns each vehicle $v$ to at most one request $M(v)$ and each request $r$ to at most one vehicle $M(r)$. That is, $M$ is always a *matching* in the bipartite graph $G$.

We focus on two main objectives:

- The *efficiency* of an assignment $M$,

$$\mathcal{E}(M) := \sum_{v \in \mathcal{V}} u_{v, M(v)} = \sum_{v \in \mathcal{V}} h_v + w_{v, M(v)}$$

- The *fairness* of an assignment $M$,

$$\mathcal{F}(M) := \min_{v \in \mathcal{V}} \{u_{v, M(v)}\} = \min_{v \in \mathcal{V}} \{h_v + w_{v, M(v)}\}$$

Let $\mathcal{M}$ be the set of all feasible assignments of instance $\mathcal{I}$, we further define the *optimal efficiency* $\mathcal{E}_{\text{opt}} := \max\{\mathcal{E}(M) \mid M \in \mathcal{M}\}$ and *optimal fairness* $\mathcal{F}_{\text{opt}} := \max\{\mathcal{F}(M) \mid M \in \mathcal{M}\}$. We will refer to the assignments that produce optimal efficiency and optimal fairness as *efficient assignment* $M_{\text{eff}}$ and *fair assignment* $M_{\text{fair}}$, respectively.

Note that our model above is flexible with respect to different features that may be of interest to the system. Below we discuss how to incorporate ridesharing (or carpooling) and how our model can handle real-time assignment with multiple time periods.

## 2.1 Ridesharing

Ridesharing refers to a ridesourcing mode in which a vehicle can serve multiple (usually no more than 2) requests simultaneously. Such a service has been provided by all major ridesourcing companies worldwide and has enormous potential for positive societal impacts in terms of pollution, energy consumption, and traffic congestion.

Our model can be easily adapted to allow ridesharing and following are the specific changes needed to be made. First, we define a *passenger* to be a past request assigned in any preceding periods that had not been picked up, or had been picked up by some vehicle and is currently en route to its destination. At any batch assignment, each vehicle $v \in \mathcal{V}$ will have its own set of passengers $S_v$. Thus, to determine whether an edge $\{v, r\} \in E$, we need to update our assignment constraints such that, for instance, the total number of occupied seats by $r$ and $p$, $\forall p \in S_v$, does not exceed the vehicle's capacity or the delay (corresponding to $r$ and each passenger $p \in S_v$) imposed by augmenting $r$ to $v$'s current route should be within some specified threshold.

Next, by allowing ridesharing, our definition of $h_v$ implies the inclusion of *trip utility* $w_{vp}$ of all passengers $p \in S_v$. It is particularly important to note the slight difference between the interpretation of *trip cost* in this setting and its single-ride counterpart.[4]

Note that all these changes only affect the structure (density) of graph $G$ and the values in $\{h_v\}_{v \in \mathcal{V}}$ associated to its edge weights. The overall model remains the same. The definition of *efficiency* and *fairness* also remains unchanged. Hence, all the results and algorithm in Section 3 directly extend to this setting. We will describe more details on the specific constraints for ridesharing relevant to our case study in Section 4.

## 2.2 Multi-Period Assignment

The above model describes a vehicle-request assignment problem in a single-batch setting. In practice, requests are collected and matched in multiple batches in real-time throughout the day. To this end, we can generalize our model to the following multi-period setting.

We split the duration of one day into $T$ discrete time periods $\{1, \dots, T\}$ (e.g., 30s per period). Requests are collected and matched during each time period. Consider a vehicle $v$ that is assigned to serve request $r$ in the current time period. In the single-ride setting, $v$ will become unavailable for $t_{vr}$ time periods while serving $r$ and reappear at the $r$'s destination afterward. In the ridesharing setting, $r$'s pickup and destination locations will be appended to the route associated to the passenger set $S_v$, as long as this satisfies the specified assignment constraints.

The definition of historical utility $h_v$ extends naturally in the context of multi-period assignment in that $h_v$ is to be updated after each batch assignment, i.e. $h_v^{t+1} = h_v^t + w_{vr}^{t+1}$. Therefore, at $t = T$, the definition of *utility*, *efficiency*, and *fairness* remain the same.

$$\iota_{vr} := \max\{0, \text{pu}(r) - \max\{t, \text{do}(p)\}\}$$

with $t$ the assignment time of $r$, $\text{pu}(r)$ the pickup time of $r$, and $\text{do}(p)$ the last dropoff time of passenger $p$ that has been on-board during the assignment of $r$. This definition caters to the case when $S_v$ is not empty and that assigning $r$ to $v$ may involve the altering of $v$'s original route in a non-trivial way. Meanwhile, in a single-ride setting, this definition just implies $r$'s pickup duration.

## 3   Efficiency-Fairness Tradeoff

In this section we analyze the efficiency and fairness tradeoff in ridesharing. Our main result is the following theorem.

**Theorem 3.1.** *Given any ridesharing problem instance $\mathcal{I}$ and any $0 \leq \lambda \leq 1$, there exists an assignment $M$ with fairness $\mathcal{F}(M) \geq \lambda \mathcal{F}_{opt}$ and efficiency $\mathcal{E}(M) \geq \frac{2}{2+\lambda}(\mathcal{E}_{opt} - n\Delta)$ simultaneously.*

Our proof is constructive. In the following we present a simple reassignment algorithm that, starting from any existing assignment $M_{\mathrm{old}}$, outputs a new assignment $M_{\mathrm{new}}$ satisfying any desired fairness threshold with bounded efficiency loss from $M_{\mathrm{old}}$.

---

**Algorithm 1:** REASSIGN $(\mathcal{I}, M_{\mathrm{old}}, f)$

---

**Input** : Instance $\mathcal{I} = \{G(\mathcal{V}, \mathcal{R}, E), \{w_{vr}\}_{\{v,r\} \in E}, \{h_v\}_{v \in \mathcal{V}}\}$,
    current assignment $M_{\mathrm{old}}$,
    fairness threshold $f \leq \mathcal{F}_{\mathrm{opt}}$.
**Output** : A new vehicle-request assignment $M_{\mathrm{new}}$

1  Compute a fair assignment $M_{\mathrm{fair}}$
2  Set $M_{\mathrm{new}} = M_{\mathrm{old}}$
3  **while** *there exists $v \in \mathcal{V}$ such that $h_v + w_{v,M_{new}(v)} < f$* **do**
4  $\quad$ $r \leftarrow M_{\mathrm{new}}(v)$
5  $\quad$ $M_{\mathrm{new}}(v) \leftarrow \emptyset$
6  $\quad$ **while** *there exists $v' \in \mathcal{V}$ such that $M_{new}(v') = M_{fair}(v)$* **do**
7  $\quad\quad$ $M_{\mathrm{new}}(v') \leftarrow \emptyset$
8  $\quad\quad$ $M_{\mathrm{new}}(v) \leftarrow M_{\mathrm{fair}}(v)$
9  $\quad\quad$ $v \leftarrow v'$
10 $\quad$ **end**
11 $\quad$ $M_{\mathrm{new}}(v) = M_{\mathrm{fair}}(v)$
12 **end**

---

Intuitively, the algorithm repeatedly chooses a vehicle $v$ whose total utility $u_{v,M_{\mathrm{new}}(v)}$ is lower than the fairness threshold $f$, and swap its assigned request to the one given out by the *fair assignment*, i.e. assign $v$ to $M_{\mathrm{fair}}(v)$. Note that this new request $M_{\mathrm{fair}}(v)$ may be assigned to another vehicle $v'$ in $M_{\mathrm{new}}$ and thus, the swapping of solution continues until no such $v'$ can be found, as described in line 6-10 of REASSIGN.

**Compute a fair assignment $M_{\mathbf{fair}}$.**   Line 1 of REASSIGN requires us to compute a fair assignment $M_{\mathrm{fair}}$. This can be done efficiently using a simple variation of the standard bipartite matching algorithm: We add $n$ *no-serve* requests $\bar{r}_1, \ldots, \bar{r}_n$ to set $\mathcal{R}$. Each $\bar{r}_i$ has only one vehicle $v_i$ connected to it with $w_{v_i, \bar{r}_i} = 0$; accordingly, we have $u_{v_i, \bar{r}_i} = h_{v_i}$. This edge represents the option of not assigning vehicle $v_i$ to any requests. Let the new request set be $\mathcal{R}^+$ and the new edge set be $E^+$. Then for any value $f$, we define $G_f := (\mathcal{V}, \mathcal{R}^+, E_f = \{\{v, r\} \in E^+ \mid h_v + w_{v,r} \geq f\})$. It is now easy to see that the optimal fairness $\mathcal{F}_{\mathrm{opt}}$ is the largest value $f$ such that $G_f$ still has a perfect matching. Such $f$ can be found via a binary search on all possible fairness thresholds. $M_{\mathrm{fair}}$ is then a perfect matching in $G_{\mathcal{F}_{\mathrm{opt}}}$.

To prove Theorem 3.1, we show a more general claim about the output of REASSIGN.

**Lemma 3.2.** *Given instance $\mathcal{I}$, current assignment $M_{old}$ and any fairness threshold $f \leq \mathcal{F}_{opt}$, algorithm REASSIGN$(\mathcal{I}, M_{old}, f)$ always outputs an assignment $M_{new}$ with fairness $\mathcal{F}(M_{new}) \geq f$ and efficiency $\mathcal{E}(M_{new}) \geq \frac{2\mathcal{F}_{opt}}{2\mathcal{F}_{opt}+f}(\mathcal{E}(M_{old}) - n\Delta)$.*

The idea of the proof of Lemma 3.2 is to consider each iteration of chain swapping (line 6-10 of REASSIGN). For some request $r$, its contribution to the decrease in efficiency is at most $\Delta$ if $r$ is matched in $M_{\mathrm{new}}$, otherwise if $r$ is 'dropped', it is bounded above by $f$. We can then bound above the number of 'dropped' requests by constructing a lower bound for $\mathcal{E}(M_{\mathrm{new}})$. The latter can be obtained from the fact that we swap the 'violating' edges with its fair counterpart (i.e. the weight of this edge is bounded below by $\mathcal{F}_{\mathrm{opt}}$) and in a non-trivial case (i.e. $\mathcal{E}_i(M_{\mathrm{new}}) \leq \mathcal{E}_i(M_{\mathrm{old}})$), we have at least 2 edges swapped in one iteration.

Finally, Theorem 3.1 can be proved directly by replacing $f$ with $\lambda F_{\text{opt}}$ in Lemma 3.2.

**Lower Bound.**  Next we focus on the theoretical lower bound for the efficiency-fairness tradeoff that any algorithm could achieve. In particular, we show that the tradeoff achieved in Theorem 3.1 is actually tight in this model.

**Theorem 3.3.** *For any $0 \leq \lambda \leq 1$ and any $\alpha$ **strictly larger** than $\frac{2}{2+\lambda}$, there always exists a problem instance $\mathcal{I}$, such that no assignment can achieve fairness $\mathcal{F}(M) \geq \lambda \mathcal{F}_{opt}$ and efficiency $\mathcal{E}(M) \geq \alpha(\mathcal{E}_{opt} - n\Delta)$ simultaneously.*

The proof uses a simple counter-example construction and is omitted.

Theorem 3.1 and 3.3 together show that among all possible algorithms that can achieve a certain fairness requirement, the efficiency achieved by our algorithm REASSIGN has the best theoretical guarantee.

## 4   Experiments

At a first glance, the theoretical guarantee obtained in Section 3 may not be enough to convince the decision maker of a ridesourcing platform to consider fairer solutions. Because the loss in efficiency, which directly translates to a revenue loss of the platform, might be too significant for fairness considerations. For example, if one wants to adopt the fairest solution, setting $\lambda = 1$ in Theorem 3.1 shows that in the worst case the platform needs to sacrifice more than $33\%$ of efficiency. However, as we will demonstrate in this section, in practice such worst case scenario will almost never happen. Through extensive experiments on real-world datasets, we show that when moving towards fairer solutions, the incurred loss in efficiency is much smaller than the theoretical prediction and in many cases negligible.

We test the performance of our algorithm in two settings: the *single-batch setting*, in which we consider all requests within a short period of time and assign them to the set of available vehicles; and the *multi-period setting*, in which the requests are collected and assigned in multiple batches in real-time. We also consider both *single-ride* and *ridesharing* setting, as described in the Preliminaries Section.

**Dataset.**   We use the publicly available dataset of taxi trips in New York City [14], which contains for each day the time and location of all of the pickups and drop-offs executed by each of the active taxis. We choose a representative 2-hour horizon, 1700 - 1900, and extract all requests originating and finishing within Manhattan, happening in May 2013. We consider the recorded pickup time as the request arrival time and the recorded passenger count as the request size. There are between 31,694 to 56,743 extracted requests each day. To reflect real road conditions and traveling time, we construct a road network of Manhattan with 3,671 nodes and 7,674 edges. For simplication purposes, we round the original pickup and drop-off location of data-extracted requests to their respective closest nodes. Travel time on each road or edge of the network is estimated based on the daily mean travel time estimate following the method in [30]. Shortest paths and travel times between all nodes are then precomputed and stored in a look-up table.

**Construction of Bipartite Graph.**    Following, we describe the specific constraints that we use in the construction of the edge set $E$ of $G = (\mathcal{V}, \mathcal{R}, E)$[5]: a vehicle $v \in \mathcal{V}$ and a request $r \in \mathcal{R}$ is connected by an edge $\{v, r\}$ iff there exists a way for $v$ to serve $r$ such that (i) the difference between $r$'s pick-up time and its request time is within a threshold $\Omega$; (ii) the total travel delay time, defined as the difference between $r$'s actual drop-off time and its earliest possible drop-off time, is within a threshold $\Gamma$; (iii) the trip utility $w_{vr} \geq 0$; (iv) if ridesharing is allowed, the total number of passengers (inclusive of $r$) on the vehicle at any time does not exceed the vehicle capacity $\chi$. For simplification purposes, we assume that any vehicles can serve up to two requests at any time.

### 4.1   Single-Batch Assignment

Our *single-batch setting* experiment aims to elicit and analyze worst-case circumstances in terms of efficiency loss. For this purpose, we construct synthetic cross-sectional scenarios (i.e. when vehicles

have been on the road for some time and are available to serve new request) by tuning the parameters that control request data extraction, vehicle positioning, and the structure of our bipartite graph (for example, by relaxing assignment contraints or changing the ratio $\frac{|V|}{|R|}$).

### 4.1.1 Experimental Setup and Data Preprocessing

We pick several days in which there are more than 200 requests arriving in the first 30 seconds of the 1700-1900 horizon and test our algorithm on each day under different scenarios. Following, we describe the exact setting that we used to produce our worst-case results. We consider all requests with trip length at least $400s$ such that we have $m = |\mathcal{R}| \in [105, 142]$. Upon initialization, we locate $n = 1.2m$ vehicles within a reasonable time-distance from the requests such that each vehicle is connected to at least 10 different requests in $G$. We define two groups of vehicles, $\mathcal{V}_H$ and $\mathcal{V}_L$, with $|\mathcal{V}_H| = m = 5|\mathcal{V}_L|$ to introduce some level of discrepancies to vehicle historical utilities and randomly generate $h_v$ such that $\forall v \in \mathcal{V}_H, h_v \sim U(200, 400)$ and $\forall v \in \mathcal{V}_L, h_v \sim U(50, 100)$. Finally, we set the maximum waiting time constraint $\Omega = 210s$, constant $c = 1$, and vehicle capacity $\chi = 4$.

### 4.1.2 Results

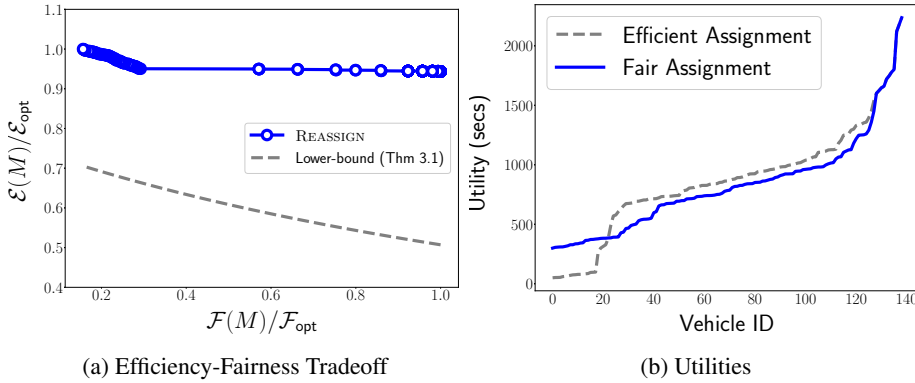

(a) Efficiency-Fairness Tradeoff             (b) Utilities

Figure 1: (a) The efficiency and fairness of the assignments output by REASSIGN with respect to different fairness thresholds, compared with the theoretical lower bound implied by Thm 3.1 (b) The utilities of all vehicles in the efficient assignment and the fair assignment, both sorted from smallest to largest.

Figure 1(a) demonstrates the tradeoff between efficiency and fairness when applying our algorithm REASSIGN with different values of fairness threshold $f$.

As one can see from the figure, when efficiency is the only concern (corresponding to the leftmost point[6]), the resulting assignment may have the lowest utility of all drivers as low as 51. However, as we start applying REASSIGN with higher and higher fairness thresholds, this lowest utility value gradually improves, until it reaches the highest point $\mathcal{F}_{opt} = 328$ in the fair solution (corresponding to the rightmost point). For the worst-off driver, the utility improvement from efficient assignment to the fair assignment is over 6-fold.

It is also important to note that, although Theorem 3.1 and 3.3 claim that the maximum efficiency loss may be as high as 33% (as indicated by the dashed curve in Figure 1(a)) in the worst case, this loss is much smaller in reality. In this example, the largest efficiency loss is less than 6% from the optimal efficiency. This is our worst-case result; the results corresponding to other days and settings preserve similar magnitudes of fairness improvement, while demonstrating even smaller efficiency losses, with many under 1%.

From our experiments, we observe three situations in which we could incur more efficiency loss: (i) when there is a denser bipartite graph, essentially more leeway to permutate between different vehicle-

request pairs; (ii) when efficient allocation has a significantly smaller overall trip cost compared to that of a fair allocation; (iii) when more requests are dropped by reassignment to a fairer solution. (i), (ii), and (iii) are intricately connected and may have competing effect up to some extent on the efficiency loss. Therefore, we can conclude that the theoretical worst-case efficiency loss does not necessarily arise even in artificial examples; here, we have seen the problem instances that show much more benign behaviour.

Finally, we compare the utilities of all vehicles (after assignment) in the *efficient* and *fair* matching. In Figure 1(b), we plot these two sets of utilities, after sorting the elements from smallest to largest. It is evident that our algorithm manages to redistribute the trip utility increments to the vehicles with low historical utilities, without sacrificing too much on the efficiency.

## 4.2 Multi-Period Assignment

We further assess the performance of our algorithm in the multi-period case, where we compute a matching assignment using REASSIGN in each period with available vehicles and requests. For this purpose, we fix $\lambda$ and set $f = \lambda \mathcal{F}_{\text{opt}}$ in REASSIGN for each period $t \in \{1, 2, \ldots, T\}$. Note that due to the dependence of the future instances on current assignments and the uncertainty inherent to future demand distributions, we cannot claim the same theoretical guarantee as shown in Section 3 at the end of the multi-period assignment horizon. Nevertheless, experiment results show that our algorithm REASSIGN performs even better in the multi-period setting, demonstrating satisfactory fairness improvement with almost no loss in efficiency.

### 4.2.1 Experimental Setup and Data Preprocessing

We discretize the 1700-1900 horizon into $T = 240$ time-steps of 30 seconds. At $t = 0$, we initialize $n = 2000$ empty vehicles with capacity $\chi = 4$ at reasonable locations based on the frequency and locations at which requests appear in the whole 2-hour horizon. Requests are generated from the dataset and collected in time windows of 30s. Each vehicle will continue to pickup and dropoff passengers following the routes assigned in batches by the central. Historical utility $h_v$ is set to 0 for all vehicles at the beginning of the simulation horizon. Then in any particular period, $h_v$ represents the accumulated trip utilities $w_{vr}$ from all the requests that vehicle $v$ has been matched to (see Section 2 for more details on the updating of $h_v$). In the case of ridesharing, $h_v$ should include the trip utilities $w_{vp}$ of all passengers $p \in S_v$. We set the maximum waiting time constraint $\Omega = 150$s and maximum delay time constraint $\Gamma = 300$s.

*Remarks.* Note that the controlling of vehicle initial locations is in line with what we did in the single-batch setting; we make sure that there are sufficiently many requests each vehicle can serve during the 2-hour horizon. With this, we want to alleviate the adverse impact that exogenous factors, such as the neighbouring structure of our network, have on fairness. Specific to the multi-period setting, we also need to take care of vehicle's intermediate locations (after each assignment). Consider the case when vehicle is assigned to a request with dropoff node having very few degrees. Due to our assumption that any vehicle stays still until it is assigned a new request, this vehicle may be stuck forever in this node. In this respect, we removed the requests whose drop-off node is not close to sufficiently many pickup nodes. We keep such removal under 4 percents of all requests in the dataset.

Following, we present the average results of our algorithm tested on 10 different days.

### 4.2.2 Results

Figure 2 shows again the efficiency and fairness tradeoff for the algorithm REASSIGN given different values of $\lambda$, in both single-ride and ridesharing setting. Compared to the single-batch setting, these results are even more extreme: there is essentially negligible efficiency loss, even for the fairest solution with $\lambda = 1$. One explanation of such phenomenon is that as time progresses, the historical utilities of all vehicles will increase to larger and larger values relative to the batch-specific trip utility increments that we can control by specifying $\lambda$'s. As a result, the analysis of Theorem 3.1 is no longer tight and correspondingly, the efficiency loss will be significantly smaller than what the theorem claims.

Figure 3 then shows the fairness improvement when we set different $\lambda$ as the parameter in REASSIGN. One can still observe a significant increase in fairness value when we shift our $\lambda$ from 0 to 1.

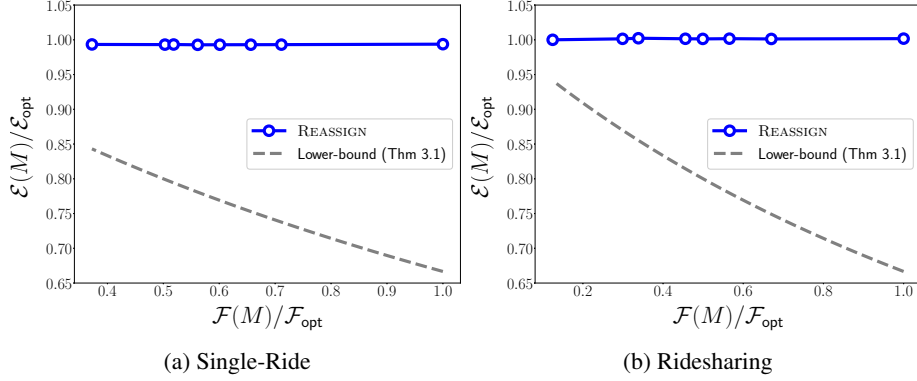

(a) Single-Ride

(b) Ridesharing

Figure 2: The efficiency and fairness of the assignments output by REASSIGN with respect to different fairness thresholds in the multi-period setting.

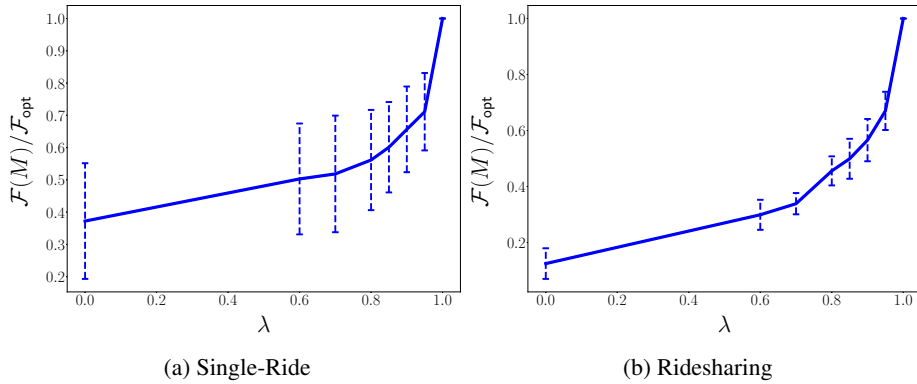

(a) Single-Ride

(b) Ridesharing

Figure 3: Relative fairness values of the assignments output by REASSIGN with regard to different $\lambda$ in the multi-period setting. The vertical intervals represent the 95% confidence intervals.

Overall, these results suggest that in practical scenarios, it is often possible to significantly improve fairness with negligible loss in system efficiency. This essentially implies that by considering fair solutions, ridesourcing enterprises will have the capacity to do much greater good without sacrificing their profitability.

## 5   Conclusion

In this paper, we deal with the problem of balancing efficiency and fairness in the context of ridesourcing request assignment. We present a simple reassignment algorithm that can compute an assignment with any desired fairness and provably good efficiency. We also provide tight upper bound on the relative efficiency loss of our solution compared to the efficient-maximizing assignment. Experiment results show that in practical scenarios, this algorithm is able to significantly improve the fairness of the assignment to drivers with very little loss on the system efficiency.

The theoretical bounds derived in our work are of independent interest and can be applied to a broader family of matching problems. How to find other suitable applications in which similar techniques or results can be applied to is one interesting future working direction. Other future research directions may include considering strictly passenger-side efficiency or different fairness criteria, such as proportional fairness [19], and measure their tradeoffs with efficiency. Finally, our investigations lead to the open question of designing a learning framework to obtain endogenously the optimal string of $\lambda$'s, that interacts with and adjusts to real-time supply-demand dynamics.

## Footnotes

[3]We make no restrictions on the structure of the set of edges $E$ and allow it to encode any physical or performance-related constraints, such as that request waiting time should be within some threshold, or vehicle type (e.g. regular, luxury) should match the request type.

[4]Generally, we define *trip cost* as follows,

[5]These are the same set of rules used in [1].

[6]Note that the threshold $f = 0$ corresponding to no fairness constraint is not necessarily binding in the reassignment procedure. For instance, in this particular case, we can keep increasing $f$ up to $.16\mathcal{F}_{opt}$ before the algorithm outputs a matching with different (and higher) fairness value.

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
