[Supplementary Material]

# Supplementary material of "Balancing Efficiency and Fairness in On-Demand Ridesourcing"

## 1 Proof of Lemma 3.2

*Proof.* Based on the reassignment procedure described in REASSIGN, the vehicles $\mathcal{V}$ can be divided into several subsets $\mathcal{S} = \{S_1, S_2, \ldots, S_t\}$, where $S_i$ consists of all vehicles that participate in the chain swapping (line 6-10 of REASSIGN) in one iteration. We assume $\mathcal{S}$ is nonempty, since otherwise we have $M_{\text{new}} = M_{\text{old}}$. Note that (1) if a vehicle $v$ appears in $S_i$ in some iteration, it will be assigned to $M_{\text{fair}}(v)$ after that iteration and will never appear again in $S_j$ for any $j > i$. Hence we have $S_i \cap S_j = \emptyset$ for any $i \neq j$; (2) there might be vehicles who do not participate in any swapping procedure. Hence $\bigcup_i S_i$ may not necessarily equal to $\mathcal{V}$. We define the set of vehicles $\mathcal{V}_o = \mathcal{V} \backslash \bigcup_{1 \leq i \leq t} S_i$. Note that $\forall v \in \mathcal{V}_o, w_{v,M_{\text{new}}(v)} = w_{v,M_{\text{old}}(v)}$.

We further define $p_i = |S_i|$ and $\mathcal{E}_i(M)$ as the *partial efficiency* of vehicles in $S_i$ of the assignment $M$, i.e. $\mathcal{E}_i(M) = \sum_{v \in S_i}(h_v + w_{v,M(v)})$.

We focus on an arbitrary set $S_i$. When $p_i = 1$, it trivially holds that $\mathcal{E}_i(M_{\text{new}}) \geq \mathcal{E}_i(M_{\text{old}})$. When $p_i \geq 2$, in the following we quantify how much efficiency loss occurs during the swapping.

Let us first define the set of $p_i$ vehicles $\{v_j\}_{1 \leq j \leq p_i}$ indexed based on the swapping order, such that $M_{\text{new}}(v_j) = M_{\text{old}}(v_{j+1}), 1 \leq j < p_i$. Thus, we have

$$
\begin{aligned}
\mathcal{E}_i(M_{\text{old}}) - \mathcal{E}_i(M_{\text{new}}) &= w_{v_1,M_{\text{old}}(v_1)} - w_{v_{p_i},M_{\text{new}}(v_{p_i})} + \sum_{2 \leq j \leq p_i} \left( w_{v_j,M_{\text{old}}(v_j)} - w_{v_{j-1},M_{\text{old}}(v_j)} \right) \\
&\leq w_{v_0,M_{\text{old}}(v_0)} - w_{v_{p_i-1},M_{\text{new}}(v_{p_i-1})} + (p_i - 1)\Delta \\
&\leq f + (p_i - 1)\Delta
\end{aligned}
\tag{1}
$$

Next, we know from REASSIGN that every vehicle $v$ in the swapping chain is reassigned to request $M_{\text{fair}}(v)$ in the output assignment $M_{\text{new}}$. Thus, we have

$$
\mathcal{E}_i(M_{\text{new}}) = \sum_{v \in S_i} \left( h_v + w_{v,M_{\text{new}}(v)} \right) \geq \sum_{v \in S_i} \mathcal{F}_{\text{opt}} \geq p_i \mathcal{F}_{\text{opt}}
$$

This implies

$$
\mathcal{E}(M_{\text{new}}) \geq \sum_{1 \leq i \leq |\mathcal{S}|} \mathcal{E}_i(M_{\text{new}}) \geq 2|\mathcal{S}| \cdot \mathcal{F}_{\text{opt}} \Rightarrow |\mathcal{S}| \leq \frac{\mathcal{E}(M_{\text{new}})}{2\mathcal{F}_{\text{opt}}}
\tag{2}
$$

Let us now consider all available vehicles $v \in \mathcal{V}$. For simplicity, we define the set of vehicles $\mathcal{V}_o = \mathcal{V} \backslash \bigcup_{1 \leq i \leq t} S_i$. Therefore, from equation (1) and (2), and the fact that for every $v \in \mathcal{V}_o$,

21   $w_{v,M_{\text{new}}(v)} = w_{v,M_{\text{old}}(v)}$, we have

$$
\begin{aligned}
\mathcal{E}(M_{\text{old}}) - \mathcal{E}(M_{\text{new})} &= \sum_{1 \leq i \leq |\mathcal{S}|} (\mathcal{E}_i(M_{\text{old}}) - \mathcal{E}_i(M_{\text{new}})) \\
&\leq \sum_{1 \leq i \leq |\mathcal{S}|} (f + (p_i - 1)\Delta) \\
&\leq |\mathcal{S}| \cdot f + n\Delta \\
&\leq \frac{f \cdot \mathcal{E}(M_{\text{new}})}{2\mathcal{F}_{\text{opt}}} + n\Delta
\end{aligned}
$$

22   Rearrange the terms in the last inequality and we obtain

$$
\mathcal{E}(M_{\text{new}}) \geq \frac{2\mathcal{F}_{\text{opt}}}{2\mathcal{F}_{\text{opt}} + f}(\mathcal{E}(M_{\text{old}}) - n\Delta)
$$

23   which is exactly what stated in the lemma. $\qquad\square$

## 24 Proof of Theorem 3.3

25   *Proof.* For any $0 \leq \lambda \leq 1$ and $\alpha > \frac{2}{2+\lambda}$, consider the following problem instances.

26   Here $\epsilon$ is set as $\min\{\frac{1}{2}(2 + \lambda - \frac{2}{\alpha}), \lambda\}$. Because $\alpha > \frac{2}{2+\lambda}$, this guarantees $\lambda \geq \epsilon > 0$.

27   Note that this problem instance has $\Delta = 0$. There are only two feasible assignments in this instance:

28        • the *efficient assignment* $M_{\text{eff}}$ (marked by solid lines) assigns $r_1$ to $v_1$ and $r_2$ to $v_2$ and gives
29          $\mathcal{E}_{\text{opt}} = 2 + \lambda - \epsilon$;

30        • the *fair assignment* $M_{\text{fair}}$ (marked by dashed lines) assigns $r_2$ to $v_1$ and leaves $r_1$ unmatched,
31          and gives $\mathcal{F}_{\text{opt}} = 1$.

32   Note that among these two assignments, $M_{\text{fair}}$ is the only one with fairness value at least $\lambda$, and we
33   have

$$
\frac{\mathcal{E}(M_{\text{fair}})}{\mathcal{E}_{\text{opt}} - n\Delta} = \frac{2}{2 + \lambda - \epsilon} < \frac{2}{2 + \lambda - (2 + \lambda - \frac{2}{\alpha})} = \alpha.
$$

34   Thus in this problem instance, any assignment that satisfies the fairness requirement stated in the
35   lemma cannot satisfy the efficiency requirement. $\qquad\square$