[Reviews · NeurIPS 2019]

Reviewer 1



UPDATE AFTER AUTHOR RESPONSE: I thank the authors for a very comprehensive author response, in particular an explanation of how the trade-off between efficiency and fairness relates and when E(M)/E(opt) does not converge. In light of these changes, I am changing my score to an accept. ---- The paper is a creative approach to reconciling efficiency and fairness in a graph-based setting. High level comments: (1) Figure 1 and 2 suggest that E(M) appears to find the same result as E_opt. Are there multiple matchings M with the same (or very close) overall utility and this algorithm finds the most fair one? Or do fairness and utility seem to converge always in this problem framework? Put another way, if we run max-flow with only regard for only overall utility, what level of fairness do we get without explicit constraints in the algorithm? When would we expect E(M) / E_opt to be much smaller? Can we construct synthetic examples -- or expand our inclusion criteria for data in the taxi dataset -- to find cases where the ratio doesn't plateau? (2) Ridesharing is a two-way market. In line 25, the introduction mentions that prior work focuses on passenger utility and efficiency. How are the two related? It feels remiss to restructure the problem as utility and fairness for the drivers without a discussion for how to reconcile this with earlier work from the passenger perspective. Could we augment the utility of each request in relation to the passenger? (3) What does the underlying dataset look like? Line 104 comments that \Delta is effectively very small across the entire graph. This seems to suggest that optimizing for fairness and utility will not have very different outcomes compared to optimizing for just utility. Empirically do we see this in the dataset? Intuitively, very uneven distributions of edge weights might create greater disparities between utilitarian efficiency and Rawsian fairness. The relatively benign decreases in efficiency because of the additional fairness criterion could be because of the edge weights. (4) How does this algorithm perform in runtime and against other baselines? Performing a binary search (line 173) to find a fair assignment M_fair while performing a max flow algorithm like Ford-Fulkerson or Edmonds-Karp runs in O(VE^2 log F) time. Is this what the authors did in implementation? Experimentally, was that a lot? A discussion of baselines -- starting with the pure utilitarian maximization and then related prior work -- would help contextualize the contribution of this paper. (5) There are other sources of unfairness in ridesharing than strictly the historical utility as defined in the experiment section. What about drivers who start from less populated areas and therefore have fewer edges (fewer requests near by) and higher edge weights (farther to drive)? Specific comments: (1) When implementing, would we directly use the fair assignment found in M_fair (line 167)? Is there a benefit to Algorithm 1 that I am missing? (2) What happens when |V| > |R|? Lack of resources aka requests may also be a case where fairness criterion are particular more important. (3) Should p2, line 100 be "maximimum utility difference of different vehicles [across all requests]"? (4) It took me a few readings of "Compute a fair assignment" (line 167) to realize you could choose to not assign a request, aka no-serve requests. Clarifying this earlier might make readability easier. (5) In Supplement p1, line 17, it is not immediately obvious to me why we use F_opt instead of f. Wouldn't using f give a tighter lower bound? (6) Figure 1b) What does "both sorted from smallest to largest" mean? (7) p6 line 255: "We also tested several other datasets" -> would be interested to see results in supplement. (8) Minor point, but could be helpful to clarify that E+ includes old E and R+ includes old R. At first reading, E+ and R+ seem to be only the new r^bar_i and corresponding edges.

Reviewer 2



(See improvements below.) AFTER READING REBUTTAL: The rebuttal bumped my score up a bit; thanks, authors, for addressing some concerns. I do think the relation to related literature is still a little light, as I mentioned in my review, and would appreciate seeing better placement in newer versions of the paper. Also, point (1) -- another discussion-based point -- should be addressed.

Reviewer 3



The paper considers the ridesharing setup (think Uber, Lyft for instance). Private drivers are linked with passengers through a mobile app. The authors study fairness in the allocation of a set of riders to a set of vehicles. A standard solution would try to maximise efficiency, i.e., maximise some aggregate measure of pairwise affinity between rider and vehicle (for instance, minimise total time to serve the request). The authors propose an alternative solution that instead optimises for a predetermined level of fairness (for instance, an upper bound on the time to serve any request) while still attempting to maximimise efficiency. The authors then show that the proposed algorithm provides a lower bound on efficiency which is optimal within the proposed model. Finally, the authors empirically study a public dataset of New York taxi trips, and conclude that significant improvements in fairness could b obtained with very small loss in efficiency. I think this is a good paper for a few reasons: - It is clearly written and easy to understand. - It is technically sound, and the proposed solution is very simple and directly implementable without anything more than a linear assignment solver. - It addresses a problem of great importance: machine learning objectives have historically been focused on the mean or median of a distribution, whereas here we see an attempt to improve the "worst-off" individual in an allocation problem, in line with the notion of fairness proposed by Rawls. - It provides further evidence of a very important finding I've seen elsewhere: often when you introduce fairness constraints the loss in efficiency is negligible. Whenever this is true, organisations have the capacity to do much greater good without affecting their profitability (and as such buying a very cheap insurance policy against potential future public retaliation due to unfair treatment or certain customers).

[Author Response · NeurIPS 2019]

We thank the reviewers for carefully reading our manuscript and providing constructive feedbacks. Below are some
comments and clarifications.

**Benign Loss in Efficiency**
**R1: High-level comments (3), Specific comments (3) | R3: Improvement.** Both R1 and R3 have named $\Delta$ as one
possible cause of the small efficiency loss and suggested its relation to the dataset. We would like to first clarify that
$\Delta$, by definition, is the maximum trip utility difference across all pairs of edges $(v, r)$ and $(v', r)$, corresponding to
the *same* request $r$, across all requests. This utility equals to value (trip length) – cost (cruising time). Since the same
request contributes the same trip value to utility when matched with any vehicle, our $\Delta$ directly translates to the cruising
time of vehicle obtained from the service constraints in waiting/delay time. For instance, when we set waiting time
constraint $\Omega = 120$s, the maximum difference in trip cost and thus, $\Delta$, is also 120s.

We further note that a small $\Delta$ would *not* cause the outcomes optimized for two different objectives to align completely.
Even in the extreme case, if we plug $\Delta = 0$ into Theorem 3.1 and 3.3, one could still see a clear tradeoff between the
two objectives. In other words, even when $\Delta = 0$ the solution optimized for just efficiency may still have very bad
fairness.

**R1: High-level comments (1), Specific comments (7), Improvement.** First, to address R1's questions in high-level
comments (1), we do find cases, especially in our multi-period experiments, where there are multiple matchings $M$
with same (or very close) overall utility (i.e., efficiency) but significantly different worst-off utility (i.e., fairness).
However, this does not mean that fairness and efficiency will always converge in this framework. In most cases, these
two objectives do not align with each other completely. In particular, one can *not* simply use a "max-flow with only
regard for overall utility" as R1 suggested. Fig 1a, 2a, 2b show the mildly decreasing trade-off curves. In all these
figures, the left-most point is the max-flow solution, meaning that no fairness constraint is imposed. In respective order
of single-batch, multi-period single-ride, multi-period ridesharing, the level of fairness we obtained from these solutions
are only 16%, 41%, and 14% of optimal fairness. Nontrivial algorithms are needed to obtain solutions with both good
efficiency and good fairness. This is exactly the purpose of Algorithm 1.

We had also conducted synthetic experiments for our single-batch setting, as R1 stated, to elicit circumstances where the
ratio $\frac{\mathcal{E}(M)}{\mathcal{E}(\text{opt})}$ doesn't plateau. From our observation, there are 3 such circumstances: (1) when there is a denser bipartite
graph or more leeway to permutate between different vehicle-request pairs, achieved by relaxing waiting/delay time
constraint $\Omega$ (thus, increasing $\Delta$) and controlling vehicle starting nodes; (2) when efficient allocation has a small overall
trip cost compare to that of a fair allocation; (3) when more requests are dropped by the reassignment to fairer solution;
this case happens very rare even in synthetic situations. We will add these discussions to the paper.

**Source of Unfairness**
**R2: Improvement (1).** We agree with R2's comments on the neighbouring structure being the potential source of
unfairness and we did try to alleviate this problem by (1) controlling the vehicle initialization node; for instance, we
sample these locations from the set of nodes that can pick-up at least 10 requests (2) removing requests that could
cause vehicles to get stuck to nodes with too few edges, while keeping the degree of removal reasonable. Both were
mentioned superficially in the paper due to space constraint. We will add these remarks to the paper.

**Static vs. Dynamic**
**R2: Improvement (2).** Our paper does have a dynamic setting component discussed in Section 2.2. Specifically, in
this multi-period setting, we allow riders and drivers to arrive and leave dynamically in rounds. We have also tested our
algorithm in this multi-period setting in the experiment section. Though this is not the main focus of this paper.

We would be happy to study and discuss the related works R2 listed in our revised version.

**Other Specific Questions**
**R1: High-level comments (5).** Our way of redistributing utility is more corrective than preventive; our algorithm will
start addressing the fairness issue once the allocated utility corresponding to these 'less-fortunate' drivers are realized.

**R1: Specific comments (1).** As we mentioned before, the fairest allocation $M_{\text{fair}}$ may have bad efficiency. REASSIGN
finds a matching $M$ that reconciliates fairness and efficiency by allowing users to choose any desired degree of fairness
through the input $\lambda$.

**R1: Specific comments (2).** We do have such results. In the single-batch experiment, we set $|\mathcal{V}| = 1.2|\mathcal{R}|$, and Fig. 1
demonstrates the result.

[Meta-Review · NeurIPS 2019]

This paper is clear, addresses a significant problem, and reviewers are favorable. Line 18: Acknowledge that this claim is controversial. Line 50: What does "mile" mean?